# Multiway Multislice PHATE: Visualizing Hidden Dynamics of RNNs through Training

## Abstract

Recurrent neural networks (RNNs) are a widely used tool for sequential data analysis, however, they are still often seen as black boxes of computation. Visualizing the internal dynamics of RNNs is a critical step in understanding the functional principles of these networks and developing ideal model architectures and optimization strategies. Previous studies typically only emphasize the network representation post-training, overlooking their evolution process throughout training. Here, we present Multiway Multislice PHATE (MM-PHATE), a novel method for visualizing the evolution of RNNs' hidden states. MM-PHATE is a graph-based embedding using structured kernels across the multiple dimensions spanned by RNNs: time, training epoch, and units. We demonstrate on various datasets that MM-PHATE uniquely preserves hidden representation community structure among units and identifies information processing and compression phases during training. The embedding allows users to look under the hood of RNNs across training and provides an intuitive and comprehensive strategy to understanding the network's internal dynamics and draw conclusions, e.g., on why and how one model outperforms another or how a specific architecture might impact an RNN's learning ability.

## 1 Introduction

Recurrent neural networks (RNNs) are artificial neural networks (ANNs) designed for sequential data. Unlike feedforward neural networks (FNNs), which treat each input as independent, RNNs model sequences of inputs to produce one or a sequence of outputs. RNNs achieve this by processing each sequence element in order while retaining a memory of past inputs through memory units (e.g., LSTMs or GRUs) or through recurrent feedback connections (Lipton et al., 2015; Kaur & Mohta, 2019). This memory effectively passes information forward in time, updating the internal state of the RNN with each input to reflect the sequence's context. RNNs are thus particularly suited for tasks where the order and relationship between elements are crucial for understanding the whole sequence, e.g., neural time-series, postural action data, etc. (Hewamalage et al., 2021).

Since their initial rise in popularity in the 1990s, researchers have created various RNN variants and training strategies to improve their training stability and ability to learn long-range time dependencies within data, such as Long-Short Term Memory (LSTM) (Hochreiter & Schmidhuber, 1996) networks and Structurally Constrained Recurrent Network (SCRN) (Mikolov et al., 2014) addressing the vanishing gradient problem (Salehinejad et al., 2018). Moreover, RNNs have a natural advantage in modeling irregular or incomplete sequential data, thanks to their flexibility in managing inputs of varying lengths. These intrinsic properties, as well as the developments of different architectures (Salehinejad et al., 2018), have led to RNNs showcasing exceptional performance across numerous domains, such as natural language processing (NLP), neuroscience and biomedical signal processing (Chen & Li, 2021; Barak, 2017; Khalifa et al., 2021). Moreover RNNs remain the state of the art for neural decoding models used in intracortical brain-computer interfaces (Deo et al., 2024).

Despite their extensive use, RNN dynamics remain poorly understood. The opaque nature of their learned representations complicates the interpretation of their performance and robustness, hinders the selection of appropriate architectures and training parameters, and slows the development of more effective models. Significant efforts have been made to address the "black-box" nature of

FNNs (Wang et al., 2021; Yu et al., 2014; Li et al., 2018). However, similar advances have not been as prevalent for RNNs, which have primarily been analyzed from the standpoint of dynamical systems analyses after training (e.g., via fixed points) (Sussillo & Barak, 2013), or by comparing the performance of different RNN architectures at the network level and investigating the role of the various components within them (Chung et al., 2014; Greff et al., 2017). Thus, new methods that facilitate the interpretation of RNNs' latent representations and their evolution during training are clearly needed.

In the field of explainable deep learning, dimensionality reduction methods are one of the more popular techniques (Karpathy et al., 2015; Hidaka & Kurita, 2017; Rauber et al., 2016; Gigante et al., 2019; Hong et al., 2020; Holtz et al., 2022). These approaches are useful as they allow researchers and engineers to readily visualize the structure of high-dimensional data, allowing them to gain intuition on structures and interactions within the data quickly. Nevertheless, they may inadvertently obscure meaningful data relationships by either emphasizing local (e.g., t-SNE Maaten & Hinton (2008)) or global (e.g., PCA Maćkiewicz & Ratajczak (1993) or Isomap Tenenbaum et al. (2000)) structures, or possess sensitivity to noise and outliers (Moon et al., 2019). These challenges are compounded in cases such as RNNs where structures along multiple dimensions (time, epoch, unit) must all be preserved to produce summaries that capture the data's complexity. As a result, traditional dimensionality reduction techniques fall short in faithfully visualizing learning in RNNs.

To study learning in FNNs, Gigante et al. (Gigante et al., 2019) introduced Multislice PHATE (M-PHATE) to visualize the network's hidden state's representations during training. M-PHATE combines a multislice graph representation to model the community structure and temporal relationship between hidden units over epochs, with Potential of Heat Diffusion for Affinity-based Transition Embedding (PHATE) (Moon et al., 2019) for dimensionality reduction. Each slice in the multi-slice graph captures the network's state at a specific epoch during training. This collection of graphs represents the dynamical system that governs the evolution of the network's hidden states, where each hidden unit is connected to itself across epochs. While M-PHATE requires only the activations of hidden units during training, the generated visualization captures key properties of network performance, such as test error accuracy, without any held-out validation data.

Despite its numerous advantages, M-PHATE is designed to visualize the evolution of FNNs across epochs but fails to account for the recurrent and sequential nature of RNNs. Research has shown that hidden states from all time-steps are crucial for RNNs' representations (Su & Shlizerman, 2020). Therefore, to fully understand the evolution of these representations, it is essential to consider network dynamics across both time-steps and epochs. To address this need, we propose Multiway Multislice PHATE (MM-PHATE). Our method captures the latent dynamics of RNNs by visualizing hidden states across both time-steps and training epochs, providing deeper insights into the complex learning processes of RNNs. Our findings indicate that MM-PHATE retains considerably more dynamic details necessary for understanding RNNs' performance compared to PCA, t-SNE, Isomap, and M-PHATE.

Our main contributions are as follows:

- We present MM-PHATE, a novel framework for visualizing the hidden dynamics of RNNs across both time-steps and epochs simultaneously, providing a new perspective to RNN's leaning trajectory, learned representation, and model performance.
- We demonstrate that MM-PHATE uniquely preserves the hidden representation community structure of hidden units throughout training by tracking each unit's learning trajectory and the correlations among their activations.
- Applying MM-PHATE to RNN dynamics identifies phases of information processing and compression during learning, an observation that aligns with information bottleneck theory.

## 2 RELATED WORK

We group existing methods to interpret RNNs into two categories: 1) performance-oriented and 2) application-oriented post-training analysis.

Performance-oriented analyses include investigating the role of components within various RNN architectures, as well as comparing different architectures and training parameters based on their

network-level performance post-training. For example, Chung et al.Chung et al. (2014) conducted a performance evaluation comparing gated RNNs, particularly GRUs and LSTMs. Similarly, Greff et al. (2017) carried out a comprehensive analysis of LSTM components. While these studies interpret the performance of different RNNs at the network level, they did not delve deeply into analyzing their hidden states, thus providing limited insight into the nature of the representations learned by these networks.

In contrast, application-oriented analyses of trained networks often involve visualizing and interpreting activation maps that depict the representations learned by hidden units after training, often in the context of a specific task or network configuration. Numerous studies have applied this approach within the field of NLP. For instance, Karpathy et al. overlaid activation heat maps over texts, demonstrating that certain units developed interpretable representations, such as tracking text length, new line beginnings, and quote initiations (Karpathy et al., 2015). Li et al. used saliency heat maps to identify words critical to the learned representations (Li et al., 2016). Strobelt et al. and Ming et al. created interactive visualizations to correlate hidden state patterns with phrases in texts (Strobelt et al., 2018; Ming et al., 2017). Beyond NLP applications, some studies explore other domains, including speech recognition (Tang et al., 2017), earth sciences (Titos et al., 2022)), and medical states (Kwon et al., 2019). While these studies provide intuitive insights into the representations learned by the networks post-training, they are tailored for a particular task and may not generalize to other tasks. Other studies tailored to understand general RNN properties include applying Proper Orthogonal Decomposition (POD) to the internal states of encoder and decoder units in Seq2Seq RNNs (Su & Shlizerman, 2020) or employing PCA to visualize the activation of recurrent units and its link to generalization(Farrell et al., 2022).

While these application-oriented studies have identified critical aspects of RNNs' hidden representation, they largely overlook the network's learning trajectory over training epochs, a crucial process that must be understood to improve RNN-based machine learning systems. To the best of our knowledge, no existing visualizations of RNNs' hidden states interpret the hidden dynamics simultaneously across time-steps and epochs.

## 3 BACKGROUND

**PHATE:** PHATE is a recent visualization technique that can capture both the local and global structure of data using diffusion processes (Moon et al., 2019). The PHATE algorithm optimizes the diffusion kernel (Coifman & Lafon, 2006) for the visualization of high dimensional data. Let $\boldsymbol{x}_i$ be a point in a high-dimensional dataset. PHATE begins by computing the Euclidean distance matrix $\boldsymbol{E}$ between all data points, where $\boldsymbol{E}_{ij} = \|\boldsymbol{x}_i - \boldsymbol{x}_j\|_2$. These distances are then transformed into affinities using an adaptive $\alpha$-decay kernel $\boldsymbol{K}_{k,\alpha}(\boldsymbol{x}_i, \boldsymbol{x}_j) = \frac{1}{2} \exp\left(-\left(\frac{\boldsymbol{E}_{ij}}{\epsilon_k(\boldsymbol{x}_i)}\right)^\alpha\right) + \frac{1}{2} \exp\left(-\left(\frac{\boldsymbol{E}_{ij}}{\epsilon_k(\boldsymbol{x}_j)}\right)^\alpha\right)$, which adapts to the data density around each point and captures local information. The parameters $\epsilon_k(x_i)$ and $\epsilon_k(x_j)$ are the $k$-nearest-neighbor distance of $x_i$ and $x_j$, and $\alpha$ controls the decay rate. The affinities are then row-normalized to obtain the diffusion operator $\boldsymbol{P} = \boldsymbol{D}^{-1}\boldsymbol{K}_{k,\alpha}$ that represents the single-step transition probabilities between data points, where $\boldsymbol{D}$ is a diagonal matrix whose entries are row sums of $\boldsymbol{K}_{k,\alpha}$. PHATE calculates the information distance between points based on their transition probabilities: $\text{dist}_{ij} = \sqrt{\|\log \boldsymbol{P}_i^t - \log \boldsymbol{P}_j^t\|^2}$, where $\boldsymbol{P}^t$ captures the transition probabilities of a diffusion process on the data over $t$ steps and $i$ and $j$ are rows in the matrix. These distances are embedded into low dimensions using Multidimensional Scaling (MDS) (Ramsay, 1966) for visualization. Local and global distances within the data's manifold are represented in PHATE by multistep diffusion probabilities. The diffusion probability of each point can capture the local context surrounding said point, allowing the construction of pairwise comparisons between all points (both neighboring and distant points) that represents the entire global context. For further details, see (Moon et al., 2019). We will use PHATE to embed RNN training dynamics, however we alter the initial graph construction to emphasize certain structures in the data we wish to visualize.

**M-PHATE:** Gigante et al. (2019) model the evolution of the hidden units in a feedforward neural network and their community structure using a multislice graph. Let $\boldsymbol{F}$ be an FNN with a total of $m$

hidden units, and let $\boldsymbol{F}^{(\tau)}$ be the representation of the network after being trained for $\tau \in \{1, ..., n\}$ epochs on the training data $\boldsymbol{X}$ sampled from a larger dataset $\boldsymbol{\Pi}$. The algorithm first calculates a shared feature space using the normalized activations of all hidden units $i \in \{1, ..., m\}$ on the input data, as a 3-dimensional tensor:

$$\boldsymbol{T}(\tau, i, k) = \frac{\boldsymbol{F}_i^{(\tau)}(\boldsymbol{Y}_k) - \frac{1}{p}\sum_\ell \boldsymbol{F}_i^{(\tau)}(\boldsymbol{Y}_\ell)}{\sqrt{\mathrm{Var}_\ell[\boldsymbol{F}_i^{(\tau)}(\boldsymbol{Y}_\ell)]}},$$

where $\boldsymbol{F}_i^{(\tau)}(\boldsymbol{Y}_k) : \mathbb{R}^d \to \mathbb{R}$ denotes the activation of the $i$-th hidden unit of $\boldsymbol{F}$ for the $k$-th sample from input data $\boldsymbol{Y}$. Here, $\boldsymbol{Y}$ is a subset of $p$ samples from the $d$-dimensional training data $\boldsymbol{X}$, with an equal number of samples from each input class, and $p \ll |\boldsymbol{X}|$. This activation tensor $\boldsymbol{T}$ is then used to calculate intraslice affinities between pairs of hidden units within an epoch $\tau$ during the training, as well as the interslice affinities between a hidden unit $i$ and itself at different epochs:

$$\boldsymbol{K}_{\mathrm{intraslice}}^{(\tau)}(i, j) = \exp\left(\frac{-\|\boldsymbol{T}(\tau, i) - \boldsymbol{T}(\tau, j)\|_2^\alpha}{\sigma_{(\tau, i)}^\alpha}\right), \boldsymbol{K}_{\mathrm{interslice}}^{(i)}(\tau, \upsilon) = \exp\left(\frac{-\|\boldsymbol{T}(\tau, i) - \boldsymbol{T}(\upsilon, i)\|_2^2}{\epsilon^2}\right)$$

where $\alpha$ is the $\alpha$-decay parameter, $\sigma_{(\tau, i)}$ is the intraslice bandwidth for unit $i$ in epoch $\tau$, and $\epsilon$ is the fixed interslice bandwidth. These matrices are combined to form an $nm \times nm$ multislice kernel matrix $\boldsymbol{K}$, which is then symmetrized, row-normalized, and visualized using PHATE in 2D or 3D.

## 4 MULTIWAY MULTISLICE PHATE

M-PHATE was shown to be a powerful tool for visualizing FNNs. However, to effectively visualize the evolution of RNNs' hidden representations, we need to consider hidden state dynamics across *time-steps* within the sequence and training epochs concurrently. In RNNs, the output from previous *time-steps* is fed as an input to current *time-steps*. This is useful in the treatment of sequences and building a memory of the previous inputs into the network. The network iteratively updates a hidden state $h$. At each *time-step* $t$, the next hidden state $h_{t+1}$ is computed using the input $x_t$ and the current hidden state $h_t$. Importantly, the network uses the same weights $W$ and biases $b$ for each *time-step*. Thus the output $y_t$ at *time-step* $t$ is $y_t = f(W \cdot h_t + b)$, where $f$ is some activation function.

Let $\boldsymbol{R}^{(\tau)}$ be the representation of an $m$-unit RNN after being trained for $\tau \in \{1, \ldots, n\}$ epochs on the training data $\boldsymbol{X} \subset \boldsymbol{\Pi}$. We denote $\boldsymbol{R}_{i,w}^{(\tau)}(\boldsymbol{Y}_k) : \mathbb{R}^d \to \mathbb{R}$ the activation of the $i$-th hidden unit of $\boldsymbol{R}$ at time-step $w \in \{1, \ldots, s\}$ in epoch $\tau$ for the $k$-th sample of $\boldsymbol{Y}$, where $\boldsymbol{Y}$ consists of $p$ samples from the training data $\boldsymbol{X}$. We construct the 4-way tensor $\boldsymbol{T}$ using the hidden unit activations as a shared feature space, which we use to calculate unit affinities across all epochs and time-steps. The tensor $\boldsymbol{T}$ is an $n \times s \times m \times p$ tensor containing the activations at each epoch $\tau \in \{1 \ldots n\}$ and time-step $w \in \{1 \ldots s\}$ of each hidden unit $\boldsymbol{R}_i$ ($i \in \{1 \ldots m\}$) with respect to each sample $\boldsymbol{Y}_k \subset \boldsymbol{X}$. To eliminate the variability in $\boldsymbol{T}$ due to the bias term $b$, we $z$-score the activation of each hidden unit at time-step $w$ and epoch $\tau$:

$$\boldsymbol{T}(\tau, \omega, i, k) = \frac{\boldsymbol{R}_{i,\omega}^{(\tau)}(\boldsymbol{Y}_k) - \frac{1}{p}\sum_\ell \boldsymbol{R}_{i,\omega}^{(\tau)}(\boldsymbol{Y}_\ell)}{\sqrt{Var_\ell[\boldsymbol{R}_{i,\omega}^{(\tau)}(\boldsymbol{Y}_\ell)]}}. \tag{1}$$

We construct a kernel over $\boldsymbol{T}$ utilizing our prior knowledge of the temporal aspect of $\boldsymbol{T}$ to capture its dynamics over epochs and time-steps. This constructed kernel, denoted $\boldsymbol{K}$, represents the weighted edges in the multislice graph of the hidden units. In this representation, each unit has two types of connections: edges between the unit to itself across epochs and time steps and, within a fixed epoch and time-step, edges between a unit and its community—the other units which have the most similar representation. The edges are weighted by the similarity in activation pattern. We define $\boldsymbol{K}$ as a $nsm \times nsm$ kernel matrix between all $m$ hidden units at all $s$ time-steps in all $n$ training epochs. The $((\tau - 1)sm + (\omega - 1)m + j)_{th}$ row or column of $\boldsymbol{K}$ refers to the $j_{th}$ unit at time-step $w$ in epoch $\tau$. We henceforth refer to the row as $\boldsymbol{K}((\tau, \omega, j), :)$ and the column as $\boldsymbol{K}(:, (\tau, \omega, j))$. In order to capture the evolution of hidden units of $R$ across time-steps and epochs, while preserving the unit's community structure, we construct a multiway multislice kernel matrix reflecting two types of connections simultaneously. Given the $\alpha$-decay parameter $\alpha$, the intrastep bandwidth for unit $i$ at time-step $w$ and epoch $\tau$: $\sigma_{(\tau, \omega, i)}$, and the fixed interstep bandwidth $\epsilon$, we define:

- Intrastep affinities between hidden units $i$ and $j$ at time-step $\omega$ in epoch $\tau$:

$$\boldsymbol{K}_{\text{intrastep}}^{(\tau,\omega)}(i,j) = \exp(-\parallel \boldsymbol{T}(\tau,\omega,i) - \boldsymbol{T}(\tau,\omega,j) \parallel_2^\alpha / \sigma_{(\tau,\omega,i)}^\alpha)$$

- Interstep affinities between a hidden unit $i$ and itself at different time-steps and epochs:

$$\boldsymbol{K}_{\text{interstep}}^{(i)}((\tau,\omega),(\eta,\nu)) = \exp(-\parallel \boldsymbol{T}(\tau,\omega,i) - \boldsymbol{T}(\eta,\nu,i) \parallel_2^2 / \epsilon^2)$$

The bandwidth $\sigma_{(\tau,\omega,i)}$ of the $\alpha$-decay kernel is set to be the distance of unit $i$ at time-step $w$ from epoch $n$ to its $k$-th nearest neighbor across units at that time-step and epoch: $\sigma_{(\tau,\omega,i)} = d_k(\boldsymbol{T}(\tau,\omega,i), \boldsymbol{T}(\tau,\omega,:))$, where $d_k(z, Z)$ denotes the $\ell_2$ distance from $z$ to its $k$-th nearest neighbor in $Z$. We used $k = 5$ in all the results presented. The use of this adaptive bandwidth means the kernel is not symmetric and thus requires symmetrization. In the interstep affinities $\boldsymbol{K}_{\text{interstep}}^{(i)}$, we use a fixed-bandwidth Gaussian kernel $\epsilon = \frac{1}{nsm} \sum_{\tau=1}^{n} \sum_{\omega=1}^{s} \sum_{i=1}^{m} d_k(\boldsymbol{T}(\tau,\omega,i), \boldsymbol{T}(:,:,i))$, the average across all time-steps in all epochs and all units of the distance of unit $i$ at time-step $t$ to its $k_{th}$ nearest neighbor among the set consisting of the same unit $i$ at all steps.

The combined kernel matrix of these two matrices contains one row and column for each unit at each time-step in each epoch, such that the intrastep affinities form a block diagonal matrix and the interstep affinities form off-diagonal blocks composed of diagonal matrices (Fig. 1a, 1b).

$$\boldsymbol{K}((\tau,\omega,i),(\eta,\nu,j)) = \begin{cases} \boldsymbol{K}_{\text{intrastep}}^{(\tau,\omega)}(i,j) & \text{if } (\tau,\omega) = (\eta,\nu) \\ \boldsymbol{K}_{\text{interstep}}^{(i)}((\tau,\omega),(\eta,\nu)) & \text{if } i = j \\ 0 & \text{otherwise} \end{cases}$$

We symmetrize this final kernel as $\boldsymbol{K}' = \frac{1}{2}(\boldsymbol{K} + \boldsymbol{K}^T)$, and row-normalize it to obtain $\boldsymbol{P} = \boldsymbol{D}^{-1}\boldsymbol{K}'$, where $\boldsymbol{D}$ is a diagonal matrix whose entries are row sums of $\boldsymbol{K}'$ and which $\boldsymbol{P}$ represents a random walk over all units at all time-steps in all epochs, where propagating from one state to another is conditional on the transition probabilities between time-step $\omega$ in epoch $\tau$ and time-step $\nu$ in epoch $\eta$. PHATE is applied to $\boldsymbol{P}$ to visualize the tensor $\boldsymbol{T}$ in two or three dimensions. This resulting visualization thus simultaneously captures information regarding the evolution of the units across both time-steps and epochs.

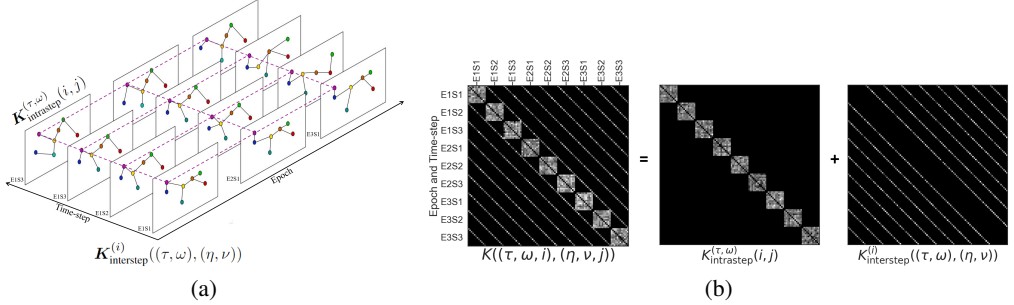

(a)  (b)

Figure 1: Example schematic of the multiway multislice graph (a) and kernel (b) used in MM-PHATE for RNNs. The intrastep kernels represent the similarities between the graph nodes at the same time-steps. The interstep kernels represent the similarities between the nodes and themselves at different time-steps and epochs.

## 5 RESULTS

We demonstrate the ability of MM-PHATE to capture useful properties of RNN learning on two datasets: 1) The Area2Bump dataset (Chowdhury et al., 2022) consisting of neural activity recorded from Brodmann's area 2 of the somatosensory cortex while macaque monkeys performed a slightly modified version of a standard center-out reaching task and 2) The Human Activity Recognition (HAR) Using Smartphones dataset (Reyes-Ortiz et al., 2012), kinematic recordings of 30 subjects performing daily living activities with a smartphone embedded with an inertial measurement unit.

## 5.1 NEURAL ACTIVITY

We begin with the Area2Bump dataset, which consists of spiking activity data from macaques (Chowdhury et al., 2022). We trained an LSTM network comprised of a single layer of 20 units to classify the direction of arm-reaching movements, and applied MM-PHATE to visualize the learning of the LSTM. In an example session where our network achieved a validation accuracy of 74%, we applied the MM-PHATE visualization to the tensor $\boldsymbol{T}$ that consisted of the network activations of all 20 units over 600 time-steps for each of 200 training epochs. In practice, we sampled time-steps and epochs to reduce memory load. Each point in the visualization represents a hidden unit at a given time-step in a given epoch (Fig. 2). Through this visualization, we observed a smooth transition of the hidden states across both time-steps and epochs, reflecting the dynamic changes in the network's internal representations throughout training. Here we compare MM-PHATE to three other dimensionality reduction techniques: PCA, t-SNE, and Isomap, and we compare to M-PHATE in the supplement (Fig. S1), using the same $\boldsymbol{T}$ tensor. We first flattened $\boldsymbol{T}$ along the epoch, time-step, and unit axis, and embedded it with each of the dimensionality reduction methods. Notably, the MM-PHATE visualization reveals a distinct split in representations during the later time-steps and epochs, highlighting unique learning patterns as the model converges—patterns that are not discernible with other methods. PCA and Isomap, while showing a seemingly smooth transition, do not show distinct differences between the early and late epochs, failing to capture how the representation transforms during learning. On the other hand, t-SNE resulted in a visualization that lacked smooth transitions across time-steps and epochs, possibly due to its sensitivity to noise.

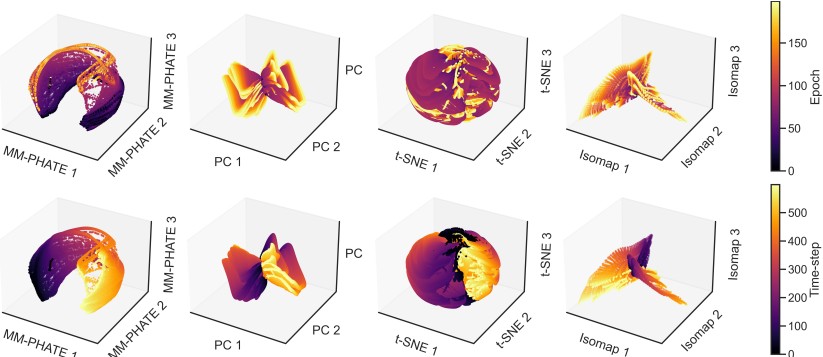

Figure 2: Area2Bump: Visualization of a 20-unit LSTM network trained for 200 epochs. The visualizations are generated using MM-PHATE, PCA, t-SNE, and Isomap, from left to right, respectively. Points are colored based on epoch (top row) or time-step (bottom-row).

### 5.1.1 INTRA-STEP ENTROPY

To evaluate the effectiveness of our embedding in capturing meaningful structures within the network's hidden dynamics, we next analyzed the flow of information during training (Fig. 3). Specifically, we aimed to analyze the spread of points in the MM-PHATE space, which corresponds to the diversity of the internal representations of the network. To quantify this property, we computed the entropy between hidden units in the embedded space at each time-step $\omega$ and compared these intra-step entropies with the accuracy and loss metrics recorded at the end of each training epoch. Conceptually, we model a general RNN trained on dataset $\boldsymbol{X}$ with label $\boldsymbol{L}$ as a Markov chain ($\boldsymbol{L} \rightarrow \boldsymbol{X} \rightarrow \boldsymbol{R}$). This allows us to compute the mutual information between $\boldsymbol{X}$ and $\boldsymbol{R}$ as $I(\boldsymbol{X}, \boldsymbol{R}) = H(\boldsymbol{R}) - H(\boldsymbol{R}|\boldsymbol{X})$, where $H(\boldsymbol{R})$ and $H(\boldsymbol{R}|\boldsymbol{X})$ are the marginal and conditional entropies. Given the deterministic nature of RNNs, $H(\boldsymbol{R}|\boldsymbol{X})$ equates to zero, indicating that $H(\boldsymbol{R})$ at each time-step $\omega$ reflects the input information the network retains during training (Tishby & Zaslavsky, 2015; Cheng et al., 2019).

Our analysis of the MM-PHATE embedding revealed a general increase in the entropy (Fig. 3) that aligns with the training epoch where the network begins to overfit (approximately epoch 100). We hypothesize that this increase may be due to memorization of noise or other nuisance variance in the training data. The observed changes in entropy at specific time-steps and epochs coincided with shifts in model performance throughout the training. This suggests that our algorithm successfully captures and retains dynamical information critical for the model's learning process. Notably, tran-

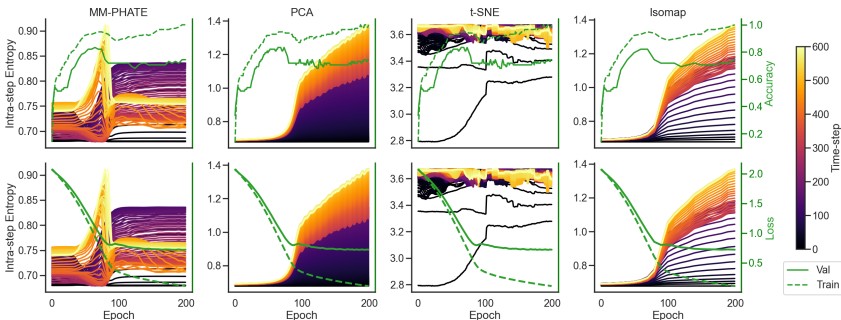

Figure 3: Intra-step entropy of all hidden units in embedding space at each time-step in each epoch, compared to training and validation accuracy (top) and losses (bottom), comparing embeddings of MM-PHATE, PCA, t-SNE, and Isomap.

sitions in validation accuracy are not always reflected in changes in the training loss curve, e.g., fluctuation of validation accuracy curve before epoch 30 cannot be observed in the training loss (Fig. 3). The MM-PHATE embedding, however, successfully identified relevant transitions coinciding with shifts in both performance measures, thus offering insights beyond those provided by accuracy and loss functions alone. Moreover, the MM-PHATE embedding reveals details on information processing through time-steps. This is not the case for the loss and accuracy metrics, which are measured once per epoch. With MM-PHATE, we obtain intra-step entropy plots for each time-step, and can thus compared dynamics across them. Notably, we observe diverging patterns between the entropy of earlier and later time steps. In this network, earlier time steps' entropy increases and plateaus around epoch 100, while later time steps mimic a spiking activity around that epoch, eventually returning back to the entropy value they had in earlier epochs. While the effects of these diverging patterns remains uncertain, MM-PHATE clearly offers additional insights into the model's dynamics.

We performed the same intra-step entropy analysis on other methods (Fig. 3). PCA failed to capture the critical dynamics associated with early subtle changes in model performance, likely due to its linear nature and emphasis on the data's global structures. Similarly, Isomap failed to capture significant dynamics related to performance changes before the onset of overfitting. Although both PCA and Isomap exhibited a rise in entropy, this increase occurred after the loss had plateaued. In contrast, MM-PHATE demonstrated an earlier rise in entropy, aligning with the transition into the plateau and thus more accurately capturing the underlying dynamics. On the other hand, t-SNE struggled to identify the increase in retained information corresponding to model overfitting and was generally inadequate in capturing the global structural dynamics evident in other techniques.

Intra-step entropy reflects how much input information the network retains during training. As the model is trained and its latent representations evolve, intra-step entropy should change accordingly. MM-PHATE clearly shows these changes in the first 100 training epochs, aligning with changes in validation accuracy, whereas traditional methods do not exhibit such responsiveness. Moreover, as the model converges, the latent representations and intra-step entropy should stabilize. In later epochs, only MM-PHATE demonstrates this desired stabilization, unlike PCA, t-SNE, and Isomap. Moreover, each time step in the input contains different information, leading the network to learn in a unique manner at each step. Consequently, intra-step entropy should vary across time steps. MM-PHATE effectively captures this variation, while PCA and Isomap show uniform entropy changes, failing to reflect the unique learning dynamics across time steps.

### 5.1.2 INTER-STEP ENTROPY

To further assess the quality of our embedding, we quantified the inter-step entropy of hidden unit activations across various time-steps $\omega$ (Fig. 4a). This metric measures the entropy between a unit's activations at different time-steps, providing insights into the temporal dynamics of each hidden unit. Our analysis identified distinct patterns among the hidden units. Notably, certain units exhibited significantly higher inter-step entropy, indicative of higher sensitivity to input changes over time and their potential role in capturing complex dependencies. These units peaked around epoch 80,

aligning with the peak in intra-step entropy at later time-steps. This suggests that these units are largely responsible for the increase in mutual information at these time-steps.

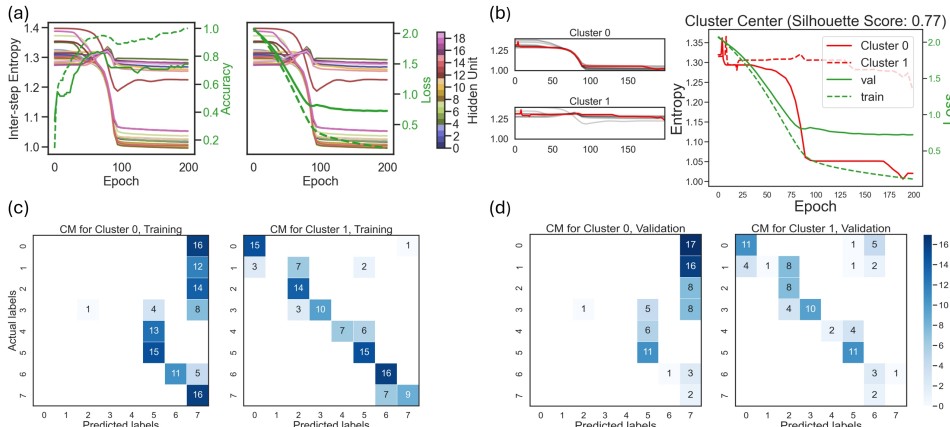

Figure 4: a) Inter-step entropy of each hidden unit across epochs, alongside model accuracy (left) and loss (right). b) Clusters of hidden units from the model's inter-step entropy trajectories across epochs. Left: Trajectories of the units in cluster 0 (12 units) and cluster 1 (8 units). Right: cluster center trajectories across epochs. c-d) Confusion matrices of clusters on training and validation data.

To validate the insights from our entropy analyses and confirm the significance of the units' distinct learning behaviors with respect to the learned representation and model performance, we clustered the hidden units into 2 groups, expecting one group to showcase the higher inter-step entropy. We then analyzed each group's specific properties. For clustering, we used k-means clustering with the Dynamic Time Warping (DTW) metric to group the hidden units into clusters based on their inter-step entropy trajectories (Fig. 4b). DTW determines an optimal match between two time series, making it more suitable as a distance metric for our sequence data. Clustering revealed distinct trends in inter-step entropy trajectories across epochs. Cluster 1 (8 units) corresponds to the units previously identified as exhibiting higher inter-step entropy, and units in cluster 0 (12 units) showed a sharp decrease in inter-step entropy around epoch 100. Following the clustering, we built new networks comprised of the post-training weighted units contained in one particular cluster, forming two sub-networks of the original network architecture to analyze their learning capabilities. By computing confusion matrices for each sub-network and comparing their predictive performance on both training and validation data, we assessed the quality of their learned representation (Fig. 4c-d). Interestingly, despite having fewer units, cluster 1 performed significantly better with both training and validation data. This outcome aligns with our previous prediction, where cluster 1 was identified as learning more complex dependencies from the input. This differential learning capability between clusters demonstrates the utility of our visualization and clustering approach in revealing critical differences in how information is processed and represented within the network.

Previous studies have indicated that the clustering property of hidden unit activation is crucial for understanding the quality of the learned representation (Su & Shlizerman, 2020; Oliva & Lago-Fernández, 2021; Ming et al., 2017). Our findings affirm the importance of capturing each unit's temporal dynamics, their community structure, as well as the evolution of these structures across time-steps and epochs.

Comparisons with other dimensionality reduction techniques highlighted their limitations (Fig. S8). PCA and Isomap failed to capture subtle dynamic variations, particularly in early epochs before the onset of overfitting. t-SNE displayed a general trend corresponding to each performance transition, however, the trajectories are noisy. More importantly, all these methods failed to differentiate individual units' learning behavior or to capture their community structure effectively. This emphasizes the superior performance of MM-PHATE in maintaining the integrity of the hidden dynamics.

In the supplemental material section, we present results for different RNN architectures, namely GRU (D.2) and Vanilla RNN (D.3), both with 20 hidden units. Using the Area2Bump dataset, we additionally tried different LSTM sizes (10, 20, 30, 40, 50) (D.4). We observed consistent visualizations when varying network size. Changing the learning rate helped confirm that the visualization

indeed reflects the model learning, i.e. the resulting change of entropies should always follow the change of model performance.

## 5.2 Analysis with Human Activity Recognition Model

We next considered an action recognition dataset. We trained 30-unit LSTM network designed for kinematics-based Human Activity Recognition (HAR) (Reyes-Ortiz et al., 2012). The model was trained for 1000 epochs, achieving a final validation accuracy of 84% (Fig. 5). We applied MM-PHATE to the tensor of hidden unit activations and repeated our analysis as in the Area2Bump dataset. Similar to the patterns observed in the Area2Bump LSTM network, the HAR network displayed an increase in intra-step entropy with the onset of overfitting. Particularly noteworthy was the gradual increase in entropy across time-steps before epoch 300. However, after this epoch, entropy at later time-steps significantly dropped, while entropy at early time-steps remained stable throughout the training epochs. This indicates a reduction in mutual information between the input and the hidden states as the model processed more input over time. This behavior aligns with findings by Farrell et al. (2022), who reported similar dynamics of information expansion and compression across time-steps in trained RNNs. Their results further elucidated how gradient-based learning mechanisms contribute to the development of robust representations by balancing these processes of expansion and compression.

Further insights were gained from the inter-step entropy analysis, which indicated that the entropy of many hidden units began to rise significantly around epoch 300, coinciding with a decrease in intra-step entropy and an improvement in model performance. As previously discussed, high inter-step entropy indicates that the units are more sensitive to changes in input across time and may be crucial for the model to learn complex dependencies. This observation also implies that despite the increase in time-dependent information retained by many hidden units, the network successfully compresses this information in its overall learned representation across epochs. Such compression seems beneficial to model performance and aligns with the principles in information bottleneck theory. This theory states that deep network learning involves a fitting phase followed by a compression phase, during which useful information is distilled from the input to enhance generalization (Cheng et al., 2019; Tishby & Zaslavsky, 2015; Butakov et al., 2023).

Building on the discussion of information compression in the HAR model, it is pertinent to question why the Area2Bump model does not exhibit a similar compression phase. Cheng et al. Cheng et al. (2019) suggest that models with insufficient generalizability often fail to demonstrate a compression phase in practice, especially when applied to complex datasets with relatively simple network architectures. Given that our models have comparable structures, we investigated the complexity of the two datasets by analyzing the number of principal components (PCs) required to explain 95% of the variance. We find that the Area2Bump data is significantly more complex than the HAR data, necessitating 35 PCs compared to only 6 for the HAR data (Fig. 6). This difference in complexity likely accounts for the absence of a noticeable compression phase in the Area2Bump model.

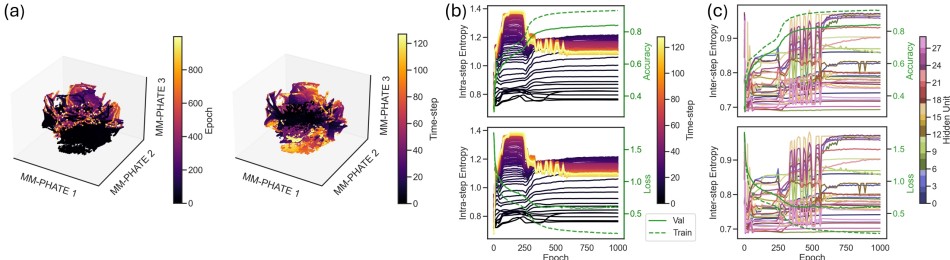

Figure 5: a) MM-PHATE visualization of a 30-unit LSTM network trained on HAR data, colored by epoch (left) and time-step (right). b) Intra-step entropy, c) and inter-step entropy.

Our analysis demonstrates that information compression occurs across both time-steps and epochs in RNNs, closely aligning with performance improvements. These results affirm the practical utility of the information bottleneck theory in RNNs and confirm that MM-PHATE effectively reveals detailed insights into the model's hidden representation and its evolution. Further research should investigate

the implications and interactions of various compression dynamics in RNNs, which could lead to more robust and generalizable network architectures.

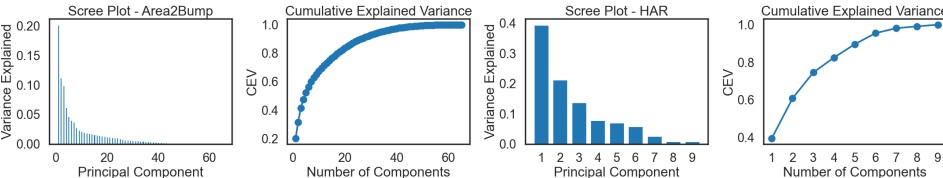

Figure 6: PCA on the 2 datasets. The Area2Bump data requires 35 PCs (left) to cover 95% of the variance. The HAR data (right) requires 6 PCs to cover 95% of variance.

## 6 CONCLUSION

In this paper, we introduced MM-PHATE, a novel dimensionality reduction technique designed to visualize the hidden dynamics of RNNs during training. MM-PHATE captures the evolution of these dynamics across both time-steps and epochs, offering insights beyond traditional metrics like accuracy and loss curves, as well as other commonly used dimensionality reduction methods. Our entropy-based analysis demonstrates that MM-PHATE can reveal distinct learning behaviors and the roles of hidden units in information flow, aligning with principles from the information bottleneck theory. This approach is especially valuable in data-limited settings, as it does not rely on external validation data. Moreover, we demonstrate the utility of analyzing the hidden state dynamics throughout the model's learning trajectory, which provides deeper insights into the internal learning processes and the evolving structure of the representation space, facilitating a more nuanced understanding of how the model captures and processes information over time.

Despite its strengths, our approach rests on several assumptions. Notably, we assume continuity in time and over epochs, as encoded in the structured graph kernel. This assumption may be violated in cases involving large learning rates, multiple significant restarts, or discontinuous activation functions, which could result in less informative visualizations. While these scenarios are not commonly explored in the literature, and PHATE has been proven to be robust against subsampling of data points, their impact should be considered (Moon et al., 2019). Additionally, while MM-PHATE is based on a computationally efficient implementation of PHATE that has shown better efficiency than traditional dimensionality reduction methods, the memory complexity of the multiway-multislice kernel introduces a bottleneck in scalability for larger architectures or datasets. Future work could address this by developing methods for sub-sampling the kernel or utilizing more memory-efficient algorithms without compromising the overall structure (Holtz et al., 2022), such as graph partitioning and merging of data points used by Kuchroo et al. (2022) in their Multiscale PHATE. Moreover, MM-PHATE does not currently leverage the internal structure of RNNs, such as attention mechanisms present in transformers (Vaswani et al., 2017). Future research will explore extending MM-PHATE to analyze transformers, which are increasingly dominant in sequential data analysis. However, RNNs remain valuable across various fields (Deo et al., 2024), especially when working with limited datasets where transformers may not be practical. Thus, while MM-PHATE is built for RNNs, its future adaptations could provide deeper insights into transformer architectures and their hidden dynamics.

### ETHICS STATEMENT

This paper presents work whose goal is to advance the field of Machine Learning. The paper has no foreseeable societal impact.

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

## A  DATASETS

We employed two datasets.

1. The Area2Bump dataset (Chowdhury et al., 2022) consists of neural activity recorded from Brodmann's area 2 of the somatosensory cortex while macaque monkeys performed a slightly modified version of a standard center-out reaching task. Dataset license: CC0 1.0. According to the authors, data was collected consistently "with the guide for the care and use of laboratory animals and approved by the institutional animal care and use committee of Northwestern University under protocol #IS00000367". This dataset, collected from monkeys thus does not contain personally identifiable information. Since it is neural data, we do not consider it to be offensive content.

   Data Statistics: The dataset includes 193 samples, split into 115 training samples and 78 testing samples. Each sample consists of 600 time steps with 65 features per time step. For the training set, the mean values across features range from 0.0001 to 0.03, with standard deviations between 0.001 and 0.02, and maximum values ranging from 0.003 to 0.12. In the test set, the mean values range from 0.00008 to 0.03, standard deviations from 0.0007 to 0.018, and maximum values from 0.0007 to 0.13.

2. The Human Activity Recognition (HAR) Using Smartphones dataset (Reyes-Ortiz et al., 2012), kinematic recordings of 30 subjects performing daily living activities with a smartphone embedded with an inertial measurement unit. Dataset license: CC BY 4.0. Information relating to participant consent was not found relating to this dataset. This dataset does not reveal participant name or identifiable information. The kinematics contained in the dataset are not considered offensive content.

   Data Details: The dataset includes six activity classes (e.g., Walking, Sitting) based on accelerometer and gyroscope data sampled at 50 Hz. The sensor data has been pre-processed with noise filtering and separated into gravitational and body motion components. Data is windowed into 2.56-second segments (128 data points) with 50% overlap, resulting in 561-dimensional feature vectors per window. The dataset is split into 70% training (21 subjects, 7352 samples) and 30% testing (9 subjects, 2947 samples).

   Data Statistics: For the training set, mean values across features range from -0.0008 to 0.8, with standard deviations between 0.1 and 0.41, and values spanning from -5.97 to 5.75. For the test set, mean values range from -0.013 to 0.8, with standard deviations between 0.095 and 0.41, and values spanning from -3.43 to 3.47.

## B  MODEL TRAINING

We used TensorFlow's Keras API for all models training and validation.

The network in Section 5.1 was trained as follows. The Area2Bump dataset was randomly split into training and validation subsets containing an equal number of samples for each input class with an 8 to 2 ratio. Additional samples that would make the training data uneven were added back to

the validation subset to make use of all samples. The network consists of an LSTM layer with 20 units. This is followed by a Flatten layer that converts the LSTM's output into a one-dimensional vector. Finally, a Dense layer with 8 units and a softmax activation function produces the output for the 8-class classification tasks. The network was trained with a batch size of 64. We used an Adam optimizer with a learning rate of $1e^{-4}$. During the training process, we recorded the activations from the LSTM layer into the activation tensor $T$ for visualization.

The network in Section 5.2 was trained as follows. The HAR dataset was preprocessed and split by the authors into training and validation subsets according to the subjects with a 7 to 3 ratio. The network consists of an LSTM layer with 30 units and a Dense layer with 6 units and a softmax activation function produces the output for the 6-class classification tasks. The network was trained with a batch size of 32. We used an Adam optimizer with a learning rate of $2e^{-5}$. During the training process, we recorded the activations from the LSTM layer into the activation tensor $T$ for visualization.

## C    IMPLEMENTATION OF VISUALIZATION METHODS

### C.1    MM-PHATE

Due to memory constraints, we only used a subset of the tensor $T$ for MM-PHATE computation. Specifically, for the Area2Bump model in section 5.1, epochs were sampled using an array combining the first 29 epochs with every 5th epoch thereafter to cover the initial rapid learning phase, and intrinsic steps were sampled using a linear space from 0 to the end (600), resulting in 100 evenly spaced steps. For the HAR model in section 5.2, we sampled the epochs using an array combining the first 29 epochs with every 10th epoch thereafter. We utilized the M-PHATE package to construct our multiway multislice graphs and for the application of PHATE.

### C.2    PCA

PCA was performed using the "PCA" class from the "sklearn.decomposition" package to reduce the dimensionality of the time trace tensor $T$—recorded during training of the Area2Bump model—to three principal components.

### C.3    T-SNE

In this analysis, we first conducted an initial dimensionality reduction on the same time trace tensor $T$ with PCA to 15 principle components, which explains 99.93% of the variance in the activations. Subsequently, t-SNE with the Barnes-Hut approximation was performed using the "sklearn.manifold.TSNE" class.

### C.4    ISOMAP

Due to memory constraints, we only used a subset of the tensor $T$ for Isomap computation. Specifically, epochs were sampled using an array combining the first 29 epochs with every 10th epoch thereafter, and intrinsic steps were sampled using a linear space from 0 to the end (600), resulting in 50 evenly spaced steps. We first conducted an initial dimensionality reduction on the sampled tensor with PCA to 15 principle components using "sklearn.decomposition.PCA" package. Then, we applied Isomap using "sklearn.manifold.Isomap" class to reduce the dimensionality to 3.

## D    ADDITIONAL EXPERIMENTS

### D.1    M-PHATE

M-PHATE was applied to the same 20-unit LSTM network trained on the Area2Bump dataset. The algorithm only incorporated the final state, or time-step, from each epoch. While the resulting visualization captures smooth transitions across epochs, it omits critical information from earlier time-steps. This loss of temporal resolution obscures insights into how the network processes input

sequences over time—an essential aspect for understanding RNNs, for instance, how much sequential input information is retained by the network.

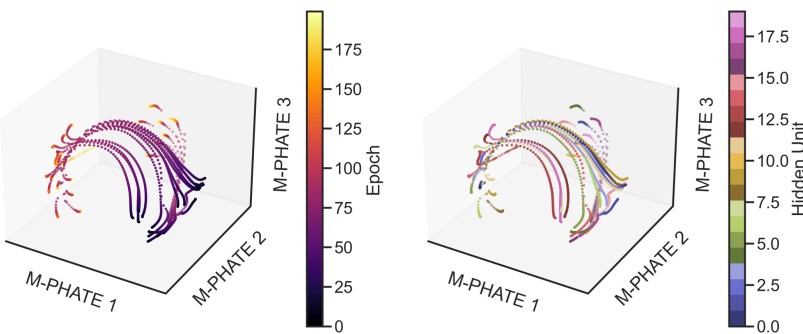

Figure S1: M-PHATE on Area2Bump LSTM: Visualization of a 20-unit LSTM network trained for 200 epochs. Each point represents a hidden unit at the last time-step in a given epoch throughout the entire training process. The points are colored based on epoch (left) or hidden unit (right).

### D.2 AREA2BUMP WITH GRU

Here is the same analysis as section 5.1 using GRU (Fig. S2, S3, S4). Other parameters were kept the same.

From these figures, it is evident that PCA and t-SNE present similar visualizations of the hidden dynamics across different network architectures, while MM-PHATE distinctly captures the unique learning behaviors of each model. Consistent with Section 5.1, PCA displays an increasing intra-step entropy even after model accuracy has plateaued, and t-SNE produces a noisy visualization. In contrast, MM-PHATE uniquely aligns its transitions well with the learning curve. Notably, the GRU model's representation appears more compact and organized compared to the LSTM model, potentially reflecting its superior performance and reduced overfitting.

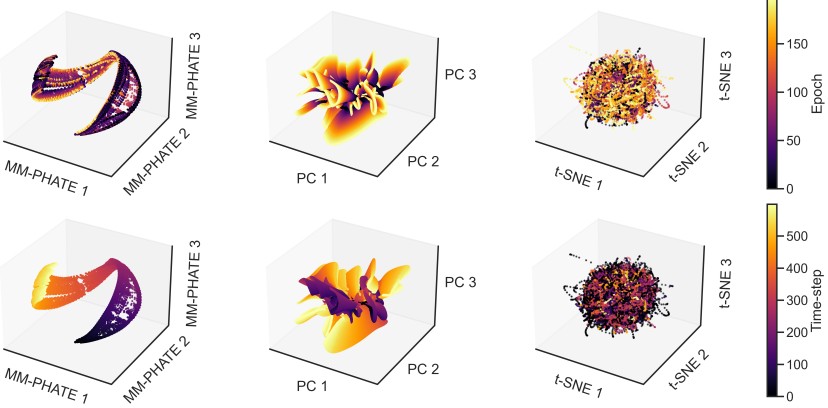

Figure S2: Area2Bump GRU: Visualization of a 20-unit GRU network trained for 200 epochs. Each point represents a hidden unit at a specific time-step in a given epoch throughout the entire training process. The visualizations are generated using MM-PHATE, PCA, and t-SNE, from left to right, respectively. Points are colored based on epoch (top row) or time-step (bottom-row)

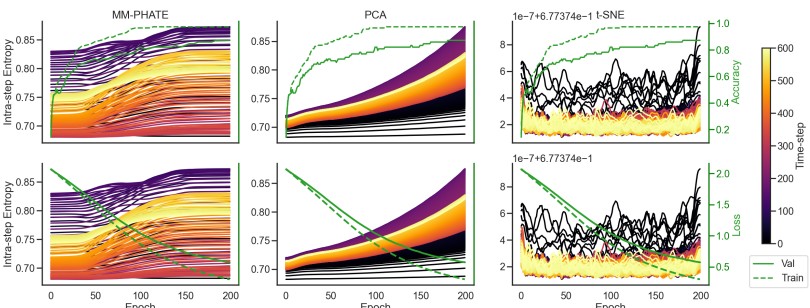

Figure S3: Area2Bump GRU: Intra-step entropy of all hidden units in embedding space at each time-step in each epoch, compared to training and validation accuracy (top) and losses (bottom), comparing embeddings of MM-PHATE, PCA, and t-SNE.

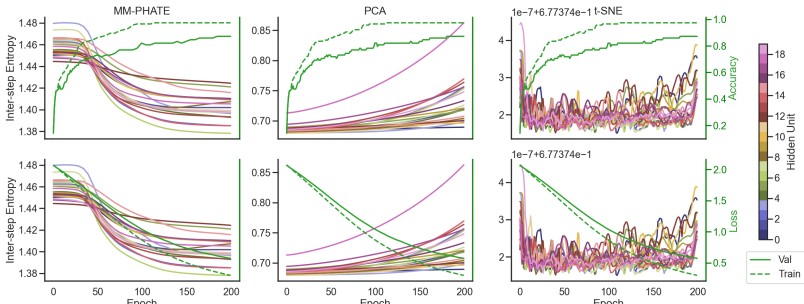

Figure S4: Area2Bump GRU: Inter-step entropy of all hidden units in embedding space of the Area2Bump model at each time-step in each epoch, compared to training and accuracies (top) and losses (bottom). From left to right, the dimensionality reduction metrics used are MM-PHATE, PCA, and t-SNE.

918
919
920
921
922
923
924
925
926
927
928
929
930
931
932
933
934
935
936
937
938
939
940
941
942
943
944
945
946
947
948
949
950
951
952
953
954
955
956
957
958
959
960
961
962
963
964
965
966
967
968
969
970
971

### D.3 Area2Bump with Vanilla RNN

Here is the same analysis as section 5.1 using vanilla RNN (Fig. S5, S6, S7). Other parameters were kept the same.

From these figures, it is evident that regardless of the network architectures, PCA exhibits a revolving pattern with overly smooth transitions across epochs, while t-SNE produces a noisy visualization. In contrast, the MM-PHATE visualization reveals that the Vanilla RNN displays a more chaotic pattern compared to the LSTM and GRU models, which is likely associated with its reduced performance and increased overfitting. Furthermore, the intra-step entropies of MM-PHATE show reduced variation across time-steps, indicating that the model struggles to process the input data effectively to generate meaningful representations.

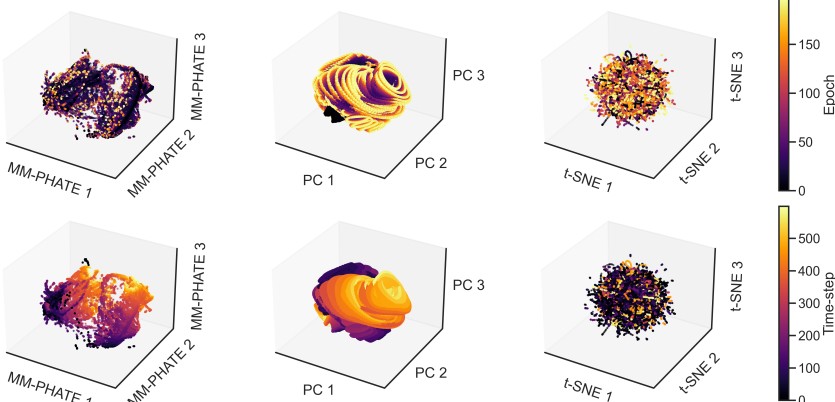

Figure S5: Area2Bump Vanilla: Visualization of a 20-unit Vanilla RNN trained for 200 epochs. Each point represents a hidden unit at a specific time-step in a given epoch throughout the entire training process. The visualizations are generated using MM-PHATE, PCA, and t-SNE, from left to right, respectively. Points are colored based on epoch (top row) or time-step (bottom-row)

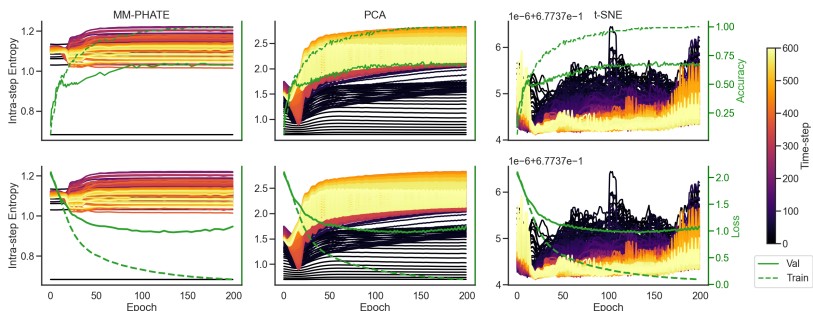

Figure S6: Area2Bump Vanilla: Intra-step entropy of all hidden units in embedding space at each time-step in each epoch, compared to training and validation accuracy (top) and losses (bottom), comparing embeddings of MM-PHATE, PCA, and t-SNE.

### D.4 Area2Bump with LSTM of Various Sizes

Here we repeat the same MM-PHATE analysis as in section 5.1 using LSTM of various sizes (Fig. S9, S10, S11). Other parameters were kept the same. These results demonstrate that MM-PHATE consistently captures smooth yet distinct transitions across epochs and time steps, regardless of the LSTM network size. The intra- and inter-step entropy analyses further reveal that these transitions closely correlate with performance changes throughout training. Specifically, we observe a general increase in intra-step entropy as models begin to overfit, suggesting that the networks increasingly memorize input information. In contrast, inter-step entropy shows a significant decline

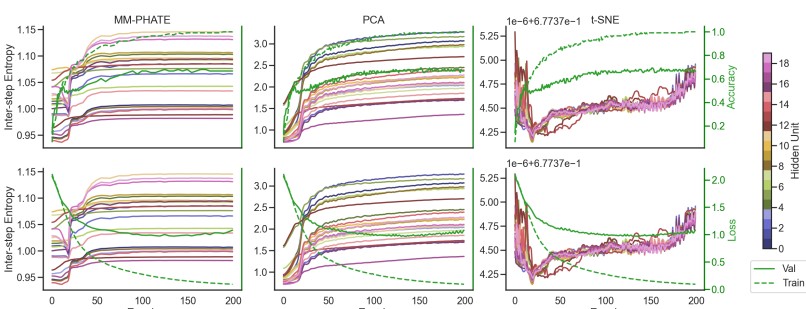

Figure S7: Area2Bump Vanilla: Inter-step entropy of all hidden units in embedding space of the Area2Bump model at each time-step in each epoch, compared to training and accuracies (top) and losses (bottom). From left to right, the dimensionality reduction metrics used are MM-PHATE, PCA, and t-SNE.

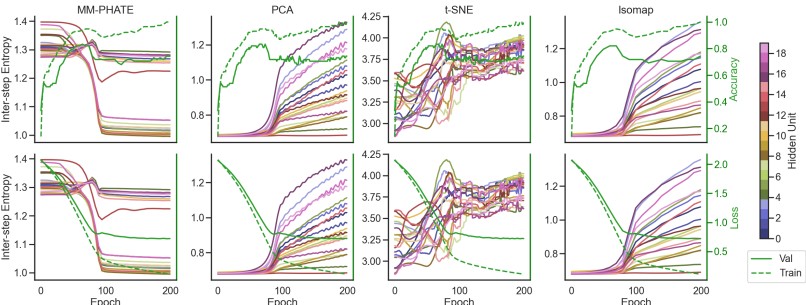

Figure S8: Area2Bump: Inter-step entropy of all hidden units in embedding space of the Area2Bump model at each time-step in each epoch, compared to training and accuracies (top) and losses (bottom). From left to right, the dimensionality reduction metrics used are MM-PHATE, PCA, t-SNE, and Isomap.

as overfitting progresses, reflecting a loss of sensitivity to input changes over time. The loss curve shows worse overfitting as the network size increases, and the networks' inter-step entropy becomes less structured.

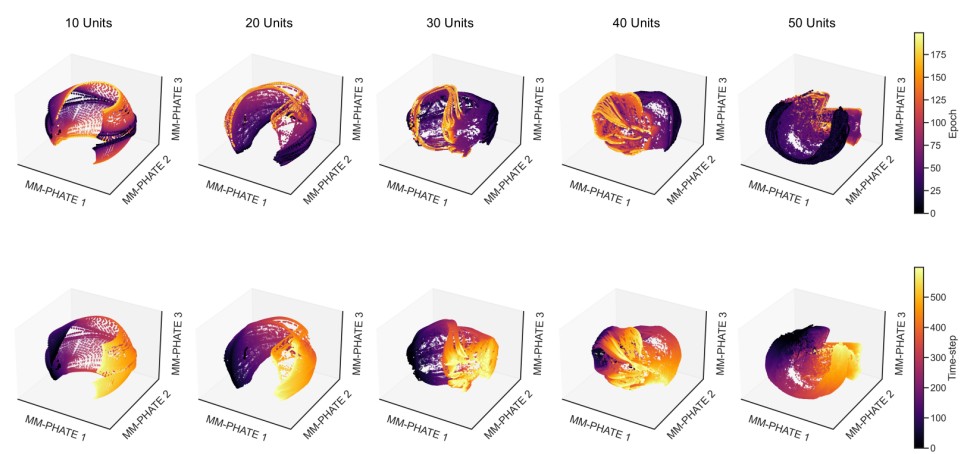

Figure S9: Area2Bump LSTM: MM-PHATE visualization of networks of size 10 to 50 (left to right). Each point represents a hidden unit at a specific time-step in a given epoch. Points are colored based on epoch (top) or timestep (bottom).

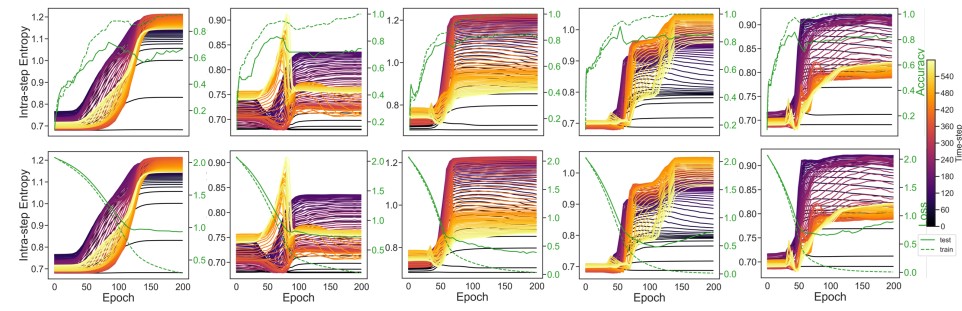

Figure S10: Area2Bump LSTM: Intra-step entropy of all hidden units in MM-PHATE embedding space at each time-step in each epoch, compared to accuracies (top) and losses (bottom). Network sizes range from 10 to 50 (left to right).

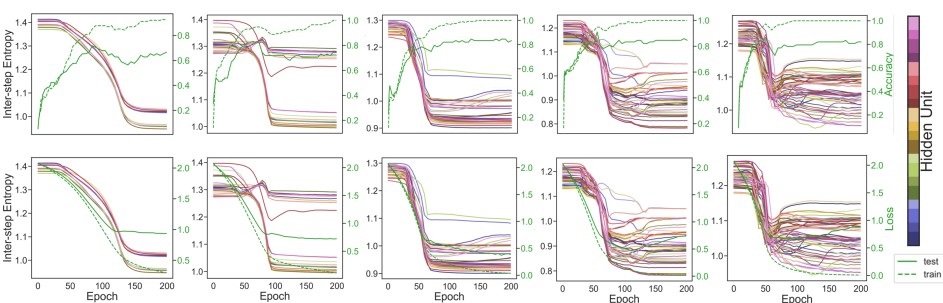

Figure S11: Area2Bump LSTM: Inter-step entropy of all hidden units in MM-PHATE embedding space of the Area2Bump model at each time-step for each unit in each epoch, compared to accuracies (top) and losses (bottom). Network sizes range from 10 to 50 (left to right).

## E    COMPUTING INFRASTRUCTURE

All but the t-SNE computation was done on a 14-core laptop running Windows 11 Home with a NVIDIA GeForce RTX 3070 Ti Laptop graphics card and 40GB of RAM. The t-SNE was done on a single 95-core internal cluster running Ubuntu 18.04.6 LTS with 10 Quadro RTX 5000 graphics cards and 755GB of RAM.

## F    MATHEMATICAL NOTATIONS

| Notation | Definition |
|---|---|
| $\boldsymbol{x}$ | Point in high dimensional space |
| $\boldsymbol{E}$ | Euclidean distance matrix between all data points $\boldsymbol{x}$ |
| $\boldsymbol{K}$ | Affinity kernel matrix |
| $\epsilon_k(x_i)$ | $k$-nearest-neighbor distance of $x_i$ |
| $\alpha$ | Parameter controlling the decay rate |
| $\boldsymbol{P}$ | Diffusion operator |
| $\boldsymbol{D}$ | Diagonal matrix of row sums of $\boldsymbol{K}$ |
| $\boldsymbol{P}^t$ | Transition probabilities of a diffusion process over $t$ steps |
| $n$ | Total number of epochs the network is trained for |
| $\boldsymbol{F}$ | Feed-forward neural network |
| $m$ | Total number of hidden units in the network |
| $\boldsymbol{X}$ | Training data, subset of $\boldsymbol{\Pi}$ |
| $\boldsymbol{\Pi}$ | Larger dataset |
| $T$ | Activation tensor |
| $\boldsymbol{Y}$ | Input data, subset of $\boldsymbol{X}$ with equal number of samples per class |
| $p$ | Number of samples in $\boldsymbol{Y}$ |
| $\boldsymbol{K}_{\text{intraslice}}^{(\tau)}(i,j)$ | Intraslice affinities between pairs of hidden units within an epoch $\tau$ |
| $\boldsymbol{K}_{\text{interslice}}^{(i)}(\tau,\upsilon)$ | Interslice affinities between a hidden unit $i$ and itself at different epochs |
| $\sigma_{(\tau,i)}$ | Intraslice bandwidth for unit $i$ in epoch $\tau$ |
| $\epsilon$ | Fixed interslice bandwidth |
| $\tau$ | Index for given epoch |
| $i,j$ | Index for given hidden unit |
| $h_t$ | RNN hidden state at time step $t$ |
| $W$ | RNN weights |
| $b$ | RNN biases |
| $y_t$ | RNN output at time step $t$ |
| $f$ | RNN activation function |
| $\boldsymbol{R}$ | Recurrent neural network |
| $w$ | Index for given time step |
| $s$ | Total number of time steps in the RNN |
| $\boldsymbol{K}_{\text{intrastep}}^{(\tau,\omega)}(i,j)$ | Intrastep affinities between hidden units $i$ and $j$ at time-step $\omega$ in epoch $\tau$ |
| $\boldsymbol{K}_{\text{interstep}}^{(i)}((\tau,\omega),(\eta,\nu))$ | Interstep affinities between a hidden unit $i$ and itself at different time-steps and epochs |
| $\sigma_{(\tau,\omega,i)}$ | Intrastep bandwidth for unit $i$ at time-step $w$ and epoch $\tau$ |
| $k$ | Number of nearest neighbors |
| $\boldsymbol{L}$ | Labels of $\boldsymbol{X}$ |

Table 1: Notations

