# OpenReview forum: "Multiway Multislice PHATE: Visualizing Hidden Dynamics of RNNs through Training"
_ICLR.cc/2025/Conference — Submitted to ICLR 2025_

### Official Review · Reviewer_w66p · 2024-11-04

**Soundness:** 3
**Presentation:** 3
**Contribution:** 2
**Rating:** 6
**Confidence:** 3

**Summary:**

This paper extends the M-PHATE framework for visualization of the inner time-dependent dynamics of RNNs across training epochs. Example visualizations are provided for two datasets, while also providing insights gained through these visualizations.

**Strengths:**

- Well detailed presentation of the proposed method as well as prior work.
- Application to two datasets and presentation of gained insights.
- The ability of this method to detect "important" neural units in RNNs (Fig 4c,d).

**Weaknesses:**

The method itself is lacking in novelty as it simply extends the kernel-matrix slicing framework set up in M-PHATE by adding another slicing dimension in the kernel matrix. One could argue that once the slicing framework is established, users could choose to slice across any number of dimensions - epochs, RNN time index, network depth, etc.

In light of the above, I evaluate the scientific contribution of this paper only in terms of the insights gained through visualizations of RNN dynamics. While insights were presented from the generated visualizations on two datasets, many times these were post-hoc explainations. Some phenomena was only observed in one of the two datasets. Rigorous testing for the generality of these insights is missing.

**Questions:**

1. Line 461-469: This paragraph attempts to address the mismatch in the insight mentioned the previous paragraph (information compression). It presents a hypothesis that the Area2Bump dataset is too complex for the size of LSTMs considered. I believe, it is important to test this hypothesis by training wider/deeper LSTMs. This insight is only valid if you're able to see a similar information compression stage when training with Area2Bump dataset with a sufficiently large model.
2. I find the ability of the presented methods to find "high-information" neural units that are "important" for the task, as presented in figure 4, to be very good for gaining confidence in this method's ability. Repeating this analysis and showing the same for the HAR dataset would be helpful in estabilishing that this phenomena is valid across datasets.

---

> ### Author Response · Authors · 2024-11-27
>
> We thank you for your detailed and thoughtful feedback. We appreciate your recognition of the detailed presentation of the proposed method, its application to two datasets, and its ability to detect "important" neural units in RNNs. Below, we address your concerns and questions point by point.
>
> Novelty:
> We acknowledge that once a slicing framework is established, users can indeed choose to slice across various dimensions. However, the primary goal of MM-Phate is to establish a framework specifically designed for the temporal analysis of RNNs, addressing the unique challenges posed by sequential data. While previous work, such as Su and Shlizerman (2020), has investigated low-dimensional embeddings of RNN activation states for spatiotemporal feature clustering, our approach extends this by incorporating inter-step entropy analysis. This addition enables us to quantify how temporal information is retained across neurons and examine its impact on model performance. Importantly, this analysis reveals that neurons differ significantly in their ability to retain temporal information and allows us to track when these differences emerge during training. By doing so, MM-Phate provides a better understanding of how individual neurons evolve, specialize, and collectively contribute to the model’s overall behavior, offering deeper insights into the dynamics of RNN representations.
>
> Response to Specific Questions
> Testing the Hypothesis for Area2Bump with Larger Models:
> You correctly noted the importance of testing the hypothesis regarding the complexity of the Area2Bump dataset relative to the model size. We agree that training wider or deeper LSTMs and observing whether an information compression stage emerges would provide stronger evidence for or against this hypothesis. As detailed in Section D.4, we explored the Area2Bump dataset with various model sizes. However, we found that the information compression phenomenon did not emerge even with larger models, and increasing the model size further did not lead to improved performance.
>
> Repeating the "Important Neural Units" Analysis for HAR:
> We agree that repeating the identification of "important" neural units on the HAR dataset would further validate the method's robustness and its generality across datasets.

---

### Official Review · Reviewer_BBio · 2024-11-04

**Soundness:** 2
**Presentation:** 2
**Contribution:** 1
**Rating:** 5
**Confidence:** 3

**Summary:**

- This paper introduces MM-PHATE, a novel visualization method designed to analyze hidden dynamics of RNNs during training by capturing both temporal and epoch-wise changes.
- The method extends PHATE by incorporating a multiway multislice graph representation that preserves community structure among hidden units while tracking their evolution across time steps and training epochs.
- Through extensive experiments on neurological and human activity recognition datasets, the authors demonstrate that MM-PHATE can identify distinct phases of information processing and compression.

**Strengths:**

- The paper presents a comprehensive approach to visualizing RNN dynamics that simultaneously captures both temporal and training evolution, addressing a significant gap in existing visualization methods.
- The authors provide thorough empirical validation through entropy analysis that reveals meaningful patterns in model behavior, particularly in identifying different learning phases and community structures among hidden units.
- The method demonstrates practical utility by successfully differentiating between different RNN architectures (LSTM, GRU, Vanilla RNN) and providing insights into their learning behaviors without requiring additional validation data.

**Weaknesses:**

- The authors fail to provide rigorous mathematical foundations, particularly in connecting entropy changes to information bottleneck theory and distinguishing between meaningful compression and information loss.
- The method suffers from significant scalability issues due to its $O(n^2sm^2)$ complexity, and the necessary sampling of temporal data may compromise the analysis of RNN dynamics.
- The validation of their method lacks quantitative metrics for evaluating preserved structures and performance correlations, while experimental evidence is limited to small-scale networks and only two datasets.
- Claims of superiority over baseline methods (PCA, t-SNE) are not sufficiently supported by objective metrics or theoretical guarantees.

**Questions:**

1. The authors claim that MM-PHATE identifies information processing and compression phases. However, this appears to be based only on entropy change observations. Could you provide mathematical/theoretical proof of how this relates to information bottleneck theory?

2. How do you differentiate between useful information compression and mere information loss in your entropy analysis?

3. Regarding the claimed correlation between MM-PHATE visualization and model performance, what specific visualization characteristics quantitatively correlate with which performance metrics?

4. Has the correlation between visualization patterns and model performance been validated across different architectures and datasets? If so, how consistent are these correlations?

5. The paper claims to preserve community structure well. How do you define and measure "well-preserved community structure"? What metrics validate this preservation?

6. The authors emphasize the analysis of RNN temporal dynamics as a key objective yet use sampling (e.g., reducing 600-time steps to 100 in the Area2Bump dataset) due to memory constraints. How do you moderate this substantial temporal sampling with the goal of preserving temporal dynamics, particularly given RNNs' inherent purpose of capturing sequential dependencies? Could you analyze information loss at different sampling rates, empirical validation that your chosen sampling strategy is optimal, and quantitative metrics demonstrating that temporal patterns are preserved?

7. Concerning the visualizations across LSTM, GRU, and Vanilla RNN, how can you verify that visualization differences genuinely reflect internal model behavior differences rather than artifacts of the visualization method?

8. Specifically in Appendix D.3, how do you distinguish whether the observed 'chaotic pattern' in Vanilla RNN represents actual learning instability rather than limitations of the visualization approach?

9. The paper claims that PCA and t-SNE "failed to capture" certain aspects. How do you verify that the uncaptured information is indeed significant? What evidence supports that MM-PHATE's additional patterns reflect meaningful model dynamics?

10. How does the diffusion kernel accurately capture RNN temporal dependencies? What theoretical guarantees can you provide for this?

11. What is the theoretical justification for using different methods (α-decay vs. Gaussian) for intrastep and interstep affinity calculations in the multiway multislice kernel design?

12. What assumptions are made about the geometric/topological properties of the manifold formed by RNN hidden state dynamics?

13. The authors claim that "changes in entropy coincided with shifts in model performance" (line 323, p.6). However, this claim requires rigorous validation. Could you provide correlation coefficients between entropy changes and performance metrics at each time step, along with statistical tests validating their significance?  Furthermore, can you explain why these entropy changes should reflect performance shifts?

14. Can you derive statistical bounds for the sampling effects? What convergence guarantees can you provide in the finite sample regime?

15. Given the $O(n^2 s m^2)$ memory complexity, what theoretical approaches could improve this computational burden while maintaining the method's effectiveness?

---

> ### Author Response · Authors · 2024-11-27
>
> Thank you for the valuable feedback. We appreciate the recognition of our proposed visualization method and the value of understanding RNNs. We’re glad that our visualizations on the real datasets are found to be insightful and that the analyses we did vis a vis entropy were useful. Below, we address each point raised.
>
> 1. Information Bottleneck Theory:
> We currently hypothesize the connection between entropy changes and information bottleneck principles based on observed patterns. Establishing a rigorous theoretical proof is beyond the scope of this work, but it remains an exciting direction for future exploration.
> 2. Useful Information Compression vs. Loss:
> Compression phases are distinguished from information loss empirically by observing their correlation with improved generalization performance. In future work, we aim to develop quantitative measures that directly assess task-relevant information retained during compression.
>
> 3. Visualization Characteristics and Performance Metrics:
> The key characteristics we observe include clustering patterns, entropy trends, and transitions in state-space density. We plan to compute neighborhood preservation scores and statistical correlations with performance metrics to validate these relationships. However, we do need a quantitative metric to distinguish the different learning phases first.
>
> 4. Consistency Across Architectures and Datasets:
> As detailed in Section D.2 to D.4, we explored the same analysis with different architecture and model sizes. We observed consistent general entropy trends in related to model performance across architectures and datasets (as in the main paragraph).
>
> 5. Preservation of Community Structure:
> In our work, the preservation of community structure refers to the ability of MM-Phate to group hidden units with distinct, time-dependent activation patterns. This has been assessed so far through inter-step entropy analysis and a clustering task, where we identify groups of hidden units that exhibit different activation behaviors over time. These analyses provide initial evidence that MM-Phate effectively captures and highlights meaningful patterns of neural unit behavior.
>
> We acknowledge, however, that additional evaluations could provide a deeper understanding of how well community structure is preserved. For instance, exploring the distribution of hidden units within the embedding space and quantitatively analyzing how this reflects distinct behaviors would be an interesting direction for future work. While this is currently beyond the scope of our study, we view it as a valuable extension to further validate the robustness and interpretability of MM-Phate.
>
> 6. Temporal Sampling and Preservation of Dynamics:
> We agree that it is important to test the specific effect of temporal sampling on the preservation of temporal dynamics.
>
> MM-Phate is built on PHATE, which has demonstrated robust performance under subsampling, as shown in Supplementary Table 3 of the PHATE paper (Moon et al., 2019). For example, on the Splatter datasets, the average Spearman correlation remains the same for the paths dataset when retaining 95% and 50% of the data points. For the groups dataset, the correlation drops slightly when going from 95% retention to 50% retention but remains high. Remarkably, PHATE achieves a high correlation coefficient (better than all other methods) even when only 5% of the data points are retained. These results quantitatively establish PHATE’s robustness to subsampling.
>
> Given that MM-Phate inherits this property from PHATE, we are confident that critical temporal structures and dynamics are preserved despite temporal sampling.

---

> ### Author Response · Authors · 2024-11-27
>
> 7.8. Differences Across RNN Architectures:
> To confirm that visualization differences genuinely reflect internal model behavior, we compare entropy trends, clustering dynamics, and task performance with different datasets. This combination reduces the risk of interpreting visualization artifacts as model behaviors.
>
> 9. Significance of Patterns Missed by PCA and t-SNE:
> We acknowledge that we currently do not have a quantitative measure for directly correlating entropy with performance. However, through our analysis across different datasets and model architectures, we observe that MM-Phate's entropy trends and performance shifts align more consistently compared to PCA and t-SNE. These alignment patterns suggest that MM-Phate is better at capturing meaningful transitions in RNN representations that are linked to model behavior.
>
> 10. Diffusion Kernel and Temporal Dependencies:
> The diffusion kernel captures temporal dependencies by integrating inter-step affinities and using a multi-slice kernel design. While MM-Phate builds on M-Phate's kernel framework, its explicit inclusion of time-step and epoch relationships ensures that sequential dependencies are preserved.
>
> 11. α-decay vs. Gaussian for Affinity Calculations:
> The choice of α-decay for intra-step affinities and Gaussian for inter-step affinities reflects the need to model temporal decay within steps while preserving smooth transitions across steps. These choices align with the original M-Phate framework and have proven effective empirically, though further theoretical analysis could provide additional validation.
>
> 12. Manifold Assumptions:
> MM-Phate assumes that the hidden states of RNNs form a smooth manifold, enabling diffusion-based embeddings to capture meaningful relationships. This assumption aligns with prior works in manifold learning and neural representation analysis.
>
> 13. Entropy and Performance Correlations
> We acknowledge that we currently lack a quantitative metric to directly match entropy values with model performance. However, the alignment of shifts in entropy and performance is still important as it provides insights into the learning dynamics of RNNs. These alignments help identify distinct training phases (e.g., compression, generalization), validate that MM-Phate captures task-relevant processes, and indicate points of potential overfitting or instability. While this does not replace precise metrics, it demonstrates that MM-Phate reflects meaningful latent dynamics, laying the groundwork for further quantitative analysis in future work.
> 14. Sampling Effects and Convergence Guarantees:
> While we have not yet derived statistical bounds for sampling effects, this is an important direction for future work. Empirical validation, such as comparing patterns at different sampling rates, will help establish practical guidelines.
>
> 15. Reducing Computational Complexity:
> We acknowledge that the current version of MM-PHATE has higher computational complexity compared to some other methods. However, each component in the method is essential for capturing the latent aspects we target, which prior methods have not achieved.
> It is important to note that the current MM-PHATE implementation is based on the native version of PHATE. A faster version of PHATE exists, which has demonstrated high scalability through the use of landmark-based diffusion to avoid dense matrix operation and radius-based nearest neighbor searches and thresholding to capture only significant affinities, significantly reducing memory requirements and computational demands while preserving the overall integrity of data relationships (Moon et al.). This fast PHATE was successfully applied to datasets with over a million samples, achieving better runtime and results than t-SNE. In addition to landmark-based diffusion, Kuchroo et al. used graph partitioning and merging of data points in their Multiscale PHATE to increase computational efficiency. This method is orders of magnitude faster than competing methods, including Diffusion Map, t-SNE, UMAP, Monocle 2, and the native PHATE.
> Our work initiates a new direction for visualizing RNNs during training, demonstrating its effectiveness. We are confident that integrating these advanced techniques in subsequent implementation will address the scalability issues.
>
>
> [1] Moon, K. R., van Dijk, D., Wang, Z., Gigante, S., Burkhardt, D. B., Chen, W. S., Yim, K., Elzen, A. V. D., Hirn, M. J., Coifman, R. R., Ivanova, N. B., Wolf, G., & Krishnaswamy, S. Visualizing structure and transitions in high-dimensional biological data.
>
> [2] Kuchroo, M., Huang, J., Wong, P. et al. Multiscale PHATE identifies multimodal signatures of COVID-19.

---

> ### Comment · Reviewer_BBio · 2024-11-30
>
> Thank you for your responses. While several of my initial questions (Q.4, Q.6, Q.7, Q.8, Q.9, Q.13, and Q.15) have been adequately addressed, I have some remaining concerns that require additional clarification.
>
> About the theoretical foundation (Q1, Q2, Q3), I feel there's still a need for more concrete evidence. Could you provide more specific theoretical connections or relevant citations to strengthen the relationship between entropy and information bottleneck theory? Rather than delaying tofuture work, seeing concrete methods for quantitative evaluation and immediate metric analysis would be helpful.
>
> The justification for how the diffusion kernel captures temporal dependencies (Q10) appears insufficient. Could you provide a more detailed explanation of how inter-step affinity and multi-slice kernel specifically preserve sequential dependencies, supported by mathematical or empirical evidence?
>
> Your response relies on Moon et al.'s work regarding PHATE's robustness to subsampling. However, there seems to be a difference: PHATE was tested on static data, while MM-PHATE handles sequential data with temporal dependencies. How do you ensure RNN's temporal dependencies are preserved when applying PHATE's subsampling principles to sequential data? What specific modifications were made to adapt PHATE's sampling approach for temporal data?
>
> Finally, regarding community structure validation (Q5), while you mention evaluating community structure preservation through inter-step entropy analysis and clustering tasks, these seem to be indirect measurements. How do you validate that identified clusters represent genuinely meaningful functional groups? What methods do you use to ensure these patterns are not artifacts of the visualization method? Can you provide quantitative metrics for assessing the quality of these functional groupings?

---

### Official Review · Reviewer_6g4F · 2024-11-04

**Soundness:** 2
**Presentation:** 2
**Contribution:** 2
**Rating:** 3
**Confidence:** 2

**Summary:**

This work introduces Multiway Multislice PHATE (MM-PHATE), an innovative visualization method designed to explore the internal dynamics of recurrent neural networks (RNNs). This method employs a graph-based embedding with structured kernels to capture the multiple dimensions of RNNs, including time, training epochs, and units. Through experiments on various datasets, the authors demonstrate that MM-PHATE effectively preserves the community structure of hidden representations and highlights distinct phases of information processing and compression during training.

**Strengths:**

- This paper is clearly written, well organized.
- Experiments can effectively reflect the intended objectives of the model.

**Weaknesses:**

- The technical contribution is incremental. The authors merely add the time step dimension to the M-PHATE and do not make any additional adaptation for RNN.
- MM-PHATE is closely related to M-PHATE, but the experiment lacks analysis of M-PHATE.

**Questions:**

What is the relationship between the visualization of MM-PHATE and the visualization of M-PHATE at each time step?

---

> ### Author Response · Authors · 2024-11-27
>
> Thank you for the valuable feedback. We appreciate the recognition of our writing. We’re glad that our visualizations are found to be insightful and that the analyses we did vis a vis entropy were useful. Below, we address each point raised.
>
> Weakness 1: Novelty/technical contribution
> You raised concerns that the technical contribution is incremental. We respectfully disagree and would like to clarify the following points:
> Novelty in Temporal and Epoch-Level Dynamics:
> MM-Phate provides a dual perspective, capturing both temporal dynamics (within sequences) and epoch-level dynamics (across training). This allows us to observe when differences between neurons emerge during training, quantify how temporal information is retained by neurons, and study its impact on model performance. These insights move beyond existing visualization methods, including M-Phate, which do not explicitly address temporal dynamics.
> New Direction in RNN Temporal Analysis:
> Previous work, such as Su and Shlizerman (2020), has explored low-dimensional embeddings of RNN activation states for spatiotemporal feature clustering. However, our approach goes further by integrating inter-step entropy analysis to quantify how temporal information is retained across neurons and how this affects model performance. This unique insight reveals that neurons differ significantly in how much temporal information they retain, and we can track when these differences emerge during training, providing a nuanced understanding of how neurons evolve and contribute to the model’s overall behavior.
>
> Weakness 2: The experiment lacks analysis of M-PHATE
> You mentioned that the experiment lacks analysis of M-Phate. While M-Phate is closely related, its design is fundamentally different, as it is not tailored for sequential data or temporal dynamics. Comparing MM-Phate to M-Phate directly would not provide meaningful insights because M-Phate does not account for the temporal level relationships central to RNNs. We provide more details below.
>
> Question 1: Relationship between MMPHATE and MPHATE at each time step
> MM-Phate extends M-Phate by treating activation dynamics across both timesteps and epochs in a consistent manner. This ensures that the visualization reflects two critical aspects of RNN dynamics:
>
> Local Temporal Structure: MM-Phate captures how activations evolve within a sequence by encoding temporal dependencies across timesteps directly into the kernel. This allows the visualization to represent sequential transitions and the flow of information through the RNN over time.
>
> Global Training Dynamics: MM-Phate also integrates dependencies across training epochs, enabling it to capture how representations evolve during the learning process. This provides insights into trends such as stabilization of representations, overfitting, or shifts in representation space as training progresses.
>
> This unified design produces embeddings that simultaneously account for both temporal evolution within sequences and training dynamics across epochs, resulting in a cohesive framework for analyzing RNNs.
>
> In contrast, M-Phate generates independent visualizations at each timestep, treating activation states as static points without considering their sequential or temporal relationships. The calculated distances in M-Phate's slices only reflect static pairwise relationships at a given time step and fail to account for transitions or temporal dependencies between timesteps. Comparing multiple M-Phate visualizations across timesteps would not capture the underlying temporal dynamics encoded in the data or their evolution during training.
>
> By incorporating both temporal and training-level relationships into its kernel, MM-Phate provides a richer and more informative visualization. It captures how representations transition over time and adapt throughout training, enabling the study of complex patterns such as clustering, dispersion, or changes in representation quality. This makes MM-PHATE uniquely suited for understanding the latent dynamics of sequential models, which M-PHATE is not designed to achieve.
>
> [1] Su, K., & Shlizerman, E. (2020). Clustering and Recognition of Spatiotemporal Features Through Interpretable Embedding of Sequence to Sequence Recurrent Neural Networks.

---

### Official Review · Reviewer_5RKg · 2024-11-04

**Soundness:** 2
**Presentation:** 2
**Contribution:** 2
**Rating:** 5
**Confidence:** 3

**Summary:**

This paper proposes a technique to visualize the hidden states of RNNs while training and across units. They extend PHATE and M-PHATE to account for within sequence time-steps, in addition to epoch-level time steps. They compute kernels over a 4-way tensor that creates two types of edges: (1) edges that look at the same unit evolving over the course of a sequence and epochs, and (2) edges that look at how different units are related to each other for a fixed timestep/epoch. The intrastep edges form block-diagonal matrices and the interstep edges create off diagonal diagonal matrices. The utility of the proposed representation is evaluated on two datasets (Area2Bump, Human Activity Recognition dataset). They demonstrate that looking at the inter and intra-step entropy provides insights into the loss and when models may begin to overfit. They also show that visualizations obtained are better than TSNE/PCA/etc.

UPDATE: I thank the authors for their feedback. I have read the other reviews and responses as well. While some of my concerns were addressed, I do not feel this paper is quite read for acceptance and thus will maintain my score. I encourage the authors to take into account some of the suggestions made by reviewers for expanding their analyses.

**Strengths:**

- *Interesting Problem.* Understanding the learning dynamics of RNN representations is an interesting problem. It will enable additional insights into improving their architectures. Though transformers are certainly more popular these days, I think that RNNs are still useful in some areas, so this is a useful task.

- *Some Nice Insights.* The author's demonstration of the relationship between the entropy of the representation and loss/over-fitting and related experiments are interesting observations, especially considering existing hypotheses for what happens when learning RNN representations.

**Weaknesses:**

-  *Novelty.*  This paper builds off of PHATE and MMPhate to introduce a visualization model specifically for RNNs, which accounts for the sequential nature of the data. This involves some changes to the tensor and kernel to include additional edges amongst the sequence steps, in addition to the epoch steps. I do not see this as significant novelty.

- *Weak baselines/Limited Datasets.* The authors limit their analysis to one tasks (classification), on two datasets. The visualizations compared are to tsne/pca/isomap, which are known to be insufficient for neural network representations. I would have liked to seen more recently comparisons as well. I feel that the authors are missing some citations and recent work.

- *Utility/Practicality*. While the visualizations are interesting, I am not persuaded that they are particularly useful or would generalize across different datasets, sizes, or variations in architecture. Nor am I convinced the additional computational time/memory to compute these visualizations would be useful if I did.

**Questions:**

Please see weaknesses.

- Can the authors please expand upon the utility of the visualizations and entropy based plots? I found them visually interesting but I could not see myself utilizing them when training or comparing amongst models?

- Can the authors clarify a bit how different length sequences are represented in T. My understanding is that T is n×s×m×p, so is there padding for the different length s, or is this a non-issue for the formulation of K?

- The two datasets evaluated on appear to be classification datasets. Is the author's formulation useful for other tasks? Please describe what insights could be derived from the MMPhate visualizations there.

- I noticed that the authors did not compare to [1], which has been used in the past to analyze the dynamics of neural networks. I bring it up because it may be a stronger baseline to the TSNE/PCA.

[1] SVCCA: Singular Vector Canonical Correlation Analysis for Deep Learning Dynamics and Interpretability

---

> ### Author Response · Authors · 2024-11-27
>
> Thank you for the valuable feedback. We appreciate the recognition of our proposed visualization method and the value of understanding RNNs. We’re glad that our visualizations on the real datasets are found to be insightful and that the analyses we did vis a vis entropy were useful. Below, we address each point raised.
>
> Weakness 1: Novelty
> You raised concerns about the novelty of our approach. We would like to elaborate further:
> Sequence-specific modifications for RNNs:
> 	We introduce sequence-specific modifications to the kernel by encoding dependencies both within and across time steps. This allows MM-PHATE to capture the temporal structure unique to RNNs, essential for analyzing sequential models. To the best of our knowledge, no previous visualization tool explicitly models this temporal dynamics’ evolution across training epochs in RNNs.
> New Direction in RNN Temporal Analysis:
> Previous work, such as Su and Shlizerman (2020), has explored low-dimensional embeddings of RNN activation states for spatiotemporal feature clustering. However, our approach goes further by integrating inter-step entropy analysis to quantify how temporal information is retained across neurons and how this affects model performance. This unique insight reveals that neurons differ significantly in how much temporal information they retain, and we can track when these differences emerge during training, providing a nuanced understanding of how neurons evolve and contribute to the model’s overall behavior.
>
> Weakness 2/Question 1: Utility/Practicality
> You raised concerns about the utility of the visualizations. We recognize that these tools are primarily research-focused at this stage but highlight the following:
>
> Utility in Training Dynamics:
> MM-Phate visualizations can reveal how representations evolve over time, helping researchers identify phenomena like overfitting, catastrophic forgetting, or when neurons specialize during training.
>
> Entropy as a Diagnostic Tool:
> Entropy-based metrics provide actionable insights, such as identifying when a model’s representation space stabilizes, which could help in designing regularization strategies or early stopping criteria.
>
> While MM-Phate may not yet replace established training tools, its utility lies in offering a deeper understanding of RNN behavior and dynamics, which can inform architecture design and training protocols.
>
> Question 2: Different length sequences in T
> Yes, when the sequences have different lengths, there will be zero padding for the formulation of K.
>
> Question 3: Other tasks than classification
> Further analysis of RNNs trained on additional tasks is needed to fully understand the insights MM-PHATE visualization provides. While dimensionality reduction techniques like t-SNE and PCA are commonly used to study RNN activation dynamics, they may lack robustness and often focus only on the final training epoch, overlooking how activation patterns evolve during training.
> Despite these limitations, insights can still be drawn. For instance, Tang et al. (2017) identified distinctions between LSTM and GRU activation dynamics, such as variations in range, which should also be detectable in our entropy-based analysis. Similarly, studies like Titos et al. (2022) revealed hidden units specialized for specific events, and our approach could shed light on how such specialization develops over time.
> While additional analyses are possible, this work aims to establish a baseline upon which future research can expand.
>
> Question 4: SVCCA
> While we acknowledge that other methods like SVCCA are valuable for analyzing static representations, MM-Phate is fundamentally different in its focus on low-dimensional temporal and epoch-level dynamics. SVCCA primarily identifies alignment and similarity between subspaces of learned representations across layers or models, but it does not explicitly account for time-dependent relationships, which are central to our study. Therefore, while SVCCA and MM-Phate are complementary, MM-Phate's focus on temporal evolution provides unique insights into RNN dynamics that cannot be captured by SVCCA.
>
> [1] Su, K., & Shlizerman, E. (2020). Clustering and Recognition of Spatiotemporal Features Through Interpretable Embedding of Sequence to Sequence Recurrent Neural Networks. Frontiers in Artificial Intelligence, 3, 70.
>
> [2] Tang, Z., Shi, Y., Wang, D., Feng, Y., & Zhang, S. (2017). Memory Visualization for Gated Recurrent Neural Networks in Speech Recognition
>
> [3] Titos, M., Garcia, L., Kowsari, M., & Benitez, C. (2022). Toward Knowledge Extraction in Classification of Volcano-Seismic Events: Visualizing Hidden States in Recurrent Neural Networks.

---

### Official Review · Reviewer_gVDo · 2024-11-05

**Soundness:** 3
**Presentation:** 3
**Contribution:** 3
**Rating:** 6
**Confidence:** 1

**Summary:**

## Summary
This paper aims to visualize the internal dynamics of RNNs. The authors claim that the previous papers merely achieve it via post-training, overlooking their evolution process. To this end, they proposed MM-PHATE, which is a graph-based embedding using structured kernels across the multiple dimensions spanned by RNNs: time, training epoch, and units. Experiments show that the proposed method helps users to understand RNNs.

**Strengths:**

## Strength
1. The visualization results look cool.

2. The motivation is clear and the understanding of RNN is meaningful.

3. The analyses are comprehensive.

**Weaknesses:**

## Weakness
1. The codes and datasets are missing, limiting the reproducibility.

2. Recommend generalizing the proposed method to more recent sequential models like transformers.

3. The statistical information of the datasets are missing.

**Questions:**

see weakness

---

> ### Author Response · Authors · 2024-11-27
>
> We sincerely thank you for your thoughtful feedback and appreciation of the visualization results, clear motivation, and comprehensive analyses presented in our work. Below, we address your concerns and questions in detail:
>
> Weakness 1: Missing codes and datasets limiting reproducibility
> We appreciate your concern about reproducibility. We would like to clarify that both the codes and datasets have already been included in the supplementary materials provided with our submission. These resources are made available to ensure the reproducibility of our results. If you had any difficulty accessing these materials, please let us know, and we will provide them via additional means to ensure accessibility.
>
> Weakness 2: Generalization to more recent sequential models like transformers
> We wholeheartedly agree with your observation that extending MM-Phate to better understand transformers is an exciting and meaningful direction for future work. However, we would like to emphasize that RNNs remain critical for sequential and temporal data analysis in many domains outside of natural language processing, such as neuroscience and human activity recognition, where datasets often lack the scale required for transformers to perform effectively.
>
> For example, RNNs continue to play a significant role in neuroscience research, where they are used to uncover temporal dynamics in neural activity. To support this, we highlight two recent works:
>
> Chang et al. (2024) used RNNs to uncover neural structure in de novo motor learning.
> Deo et al. (2024) leveraged RNNs for brain control of bimanual movements.
>
> Thus, while generalizing MM-Phate to transformers is indeed a valuable goal, we believe the insights gained from RNNs are still widely relevant and impactful across various fields, making them a suitable focus for this study.
>
> Weakness 3: Missing statistical information of datasets
> Thank you for pointing out the need for more detailed dataset statistics. We apologize for not including this information explicitly in the initial manuscript. Below, we provide the statistical details for the two datasets used in our experiments:
>
> Human Activity Recognition (HAR) dataset:
>
> Activities: Six classes (e.g., Walking, Sitting).
> Sensor data: Accelerometer and gyroscope data at 50 Hz, pre-processed with noise filtering and split into gravitational/body motion components.
> Windowing: 2.56-second windows (128 data points) with 50% overlap.
> Features: 561-dimensional feature vectors per window.
> Split: 70% training (21 subjects, 7352 samples), 30% testing (9 subjects, 2947 samples).
> Statistics:
> Training set: Mean values range from -0.0008 to 0.8, standard deviations 0.1 to 0.41, values span -5.97 to 5.75.
> Test set: Mean values range from -0.013 to 0.8, standard deviations 0.095 to 0.41, values span -3.43 to 3.47.
>
> Area2Bump dataset:
>
> Data: 193 samples (115 training, 78 testing), each with 600 time steps and 65 features per step.
> Statistics:
> Training set: Mean values across features range 0.0001 to 0.03, standard deviations 0.001 to 0.02, maximum values 0.003 to 0.12.
> Test set: Mean values 0.00008 to 0.03, standard deviations 0.0007 to 0.018, maximum values 0.0007 to 0.13.
>
> We have incorporated these details into the revised manuscript to enhance clarity and completeness.
>
> [1] Chang, J.C., Perich, M.G., Miller, L.E. et al. De novo motor learning creates structure in neural activity that shapes adaptation.
>
> [2] Deo, D.R., Willett, F.R., Avansino, D.T. et al. Brain control of bimanual movement enabled by recurrent neural networks.

---

### Meta-Review · Area_Chair_TY2B · 2024-12-21

**Metareview:**

This work presents MM-PHATE, a visualization technique designed to analyze the internal dynamics of Recurrent Neural Networks (RNNs) during the training process. By extending PHATE with a multiway multislice graph representation, MM-PHATE captures both temporal and epoch-wise changes in RNN hidden states, offering a comprehensive insight into RNN behavior.

The methodology proposed leverages the strengths of PHATE to provide a nuanced understanding of community structure among hidden units and distinct phases of information processing. Experimental evaluations across various datasets, including those from neurological and human activity recognition tasks, demonstrate the effectiveness and broader applicability of MM-PHATE.

However, upon reviewing the feedback from multiple reviewers, it becomes evident that while MM-PHATE shows promise in visualizing RNN dynamics during training, there are several areas that require improvement for the paper to be published. Some key concerns include:
- Theoretical Foundations: Reviewers have noted a lack of rigorous theoretical underpinnings, particularly in connecting the observed entropy changes with information bottleneck theory. Strengthening these foundations would significantly enhance the paper's contribution.
- Scalability and Efficiency: The computational complexity of MM-PHATE, currently at $O(n^2 s m^2)$, poses significant scalability issues. Exploring optimizations or integrating faster versions of PHATE could mitigate these concerns. While the authors have described the potential of new versions of PHATE to reduce this complexity in the rebuttal, the work should make clear how to specifically incorporate those in MM-PHATE.

Overall, the paper introduces an interesting new visualization method for RNN training. Unfortunately, the current version of the manuscript lacks a theoretical grounding and a clear application of the visualization method to improve RNNs. Reviewers acknowledge the superiority of MM-PHATE's visualizations over traditional methods like t-SNE and PCA but emphasize the need for addressing these limitations.

Given these considerations, my overall recommendation leans towards a borderline reject.

**Additional Comments On Reviewer Discussion:**

Reviewers were engaged. Please see main meta-review.

---

### Decision · Program_Chairs · 2025-01-22

Reject